# Follow-up of clinical and sonographic features after extracorporeal shock wave therapy in painful plantar fibromatosis

**Jin Tae Hwang**[1], **Kyung Jae Yoon**[1], **Chul-Hyun Park**[1], **Jae Hyeoung Choi**[1], **Hee-Jin Park**[2], **Young Sook Park**[3], **Yong-Taek Lee**[1] *

**1** Department of Physical & Rehabilitation Medicine, Kangbuk Samsung Hospital, Sungkyunkwan University School of Medicine, Seoul, Republic of Korea, **2** Department of Radiology, Kangbuk Samsung Hospital, Sungkyunkwan University School of Medicine, Seoul, Republic of Korea, **3** Department of Physical & Rehabilitation Medicine, Samsung Changwon Hospital, Sungkyunkwan University School of Medicine, Seoul, Republic of Korea

* yongtaek1.lee@gmail.com

## Abstract

### Background

Extracorporeal shock wave therapy (ESWT) has been used as a safe alternative treatment for refractory musculoskeletal diseases, such as plantar fasciitis, Achilles tendinopathy and gluteal tendinopathy, and various forms of fibromatosis including palmar or penile fibromatosis. However, there is limited published data for clinical and sonographic features of plantar fibromatosis after ESWT. The purpose of this study was to evaluate the long-term clinical outcome of ESWT in ultrasonography-confirmed plantar fibromatosis and ultrasonographic changes of plantar fibroma after ESWT.

### Methods

Medical charts of 26 patients (30 feet) with plantar fibromatosis confirmed by ultrasonography were reviewed. Finally, a total of 10 feet who underwent ESWT for "Poor" or "Fair" grade of Roles-Maudsley Score (RMS) and symptoms persisted for >6 months were included in this study. Short-term follow-up was conducted one week after ESWT and long-term follow-up time averaged 34.0 months. The Numerical Rating Scale (NRS) and RMS were collected for the evaluation of clinical features. Follow-up ultrasonography was conducted at long-term follow-up and changes of plantar fibroma was assessed. A greater than 50% reduction in the NRS and achievement of a "good" or "excellent" grade in the RMS were regarded as treatment success. Additionally, medical charts of 144 patients (168 feet) with plantar fasciitis confirmed by ultrasonography were reviewed and subsequently, 42 feet who underwent ESWT with the same protocol were included for the comparison of clinical features.

### Results

In plantar fibromatosis, baseline NRS (6.2 ± 1.3) and RMS (3.5 ± 0.5) were significantly improved at short-term follow-up (NRS, 1.8 ± 1.0; RMS, 2.0 ± 0.8, *P* < .001, respectively)

**Data Availability Statement:** All relevant data are within the manuscript and its Supporting Information files.

**Funding:** The author(s) received no specific funding for this work.

**Competing interests:** The authors have declared that no competing interests exist.

and long-term follow-up (NRS, 0.6 ± 1.1; RMS, 1.4 ± 0.8, $P < .001$, respectively). Treatment success was recorded in seven feet (70.0%) at short-term follow-up and 8 feet (80%) at long-term follow-up, which is comparable to that of the plantar fasciitis group (28 feet, 66.7%; 35 feet, 83.3%, respectively). In long-term follow-up ultrasonography, mean fibroma thickness was reduced from 4.4±1.0 to 2.6±0.8 mm ($P = .003$); however, length and width were not significantly changed. There were no serious adverse effects.

## Conclusion

While these are preliminary findings, and must be confirmed in a randomized placebo control study, ESWT can have a beneficial long-term effect on pain relief and functional outcomes in painful plantar fibromatosis. However, ESWT is unlikely to affect the ultrasonographic morphology of plantar fibroma, with the exception of reducing the thickness.

## Level of evidence

Level III, retrospective cohort study.

## Introduction

Plantar fibromatosis, also known as Ledderhose disease, is an uncommon, benign, and hyper-proliferative fibrous tissue disease characterized by localized proliferation of fibrotic tissue and nodular formation in the plantar fascia. This condition usually progresses slowly and can cause pain, functional disability, and decreased quality of life. Although the etiology remains largely unknown, it is often associated with palmar and penile fibromatosis, which are known as Dupuytren's disease and Peyronie's disease, respectively [1–5].

Ultrasonography or magnetic resonance imaging can be used to confirm the diagnosis and measure the size and depth of the nodule [1–3, 6, 7]. Early stage conservative therapy includes non-steroidal anti-inflammatory drugs (NSAIDs), local corticosteroid injections, physical therapy, and custom-made insoles. Radiotherapy and surgical treatment may be considered in refractory cases [1–3]. However, radiotherapy can cause side effects such as lymphatic edema or fracture of irradiated bone, and surgery has a high recurrence rate that can range from 57% to 100% [1, 3, 8, 9]. Extracorporeal shock wave therapy (ESWT) has been used as a safe alternative treatment for chronic refractory musculoskeletal disease, such as plantar fasciitis Achilles tendinopathy and gluteal tendinopathy [10–13]. Previous studies have shown that ESWT also can be therapeutically applied to various forms of fibromatosis such as penile fibromatosis [14–19] and palmar fibromatosis [20–22] to reduce pain and soften nodules, although it did not affect the physical size of the nodules [3, 15, 16, 19, 23].

In terms of plantar fibromatosis, there were two case series that reported the pain-relieving effect of ESWT in 6 patients and 2 patients, respectively [24, 25]. However, these studies did not demonstrate the diagnostic process of plantar fibromatosis. In addition, there is still limited published data for clinical and sonographic features of plantar fibromatosis after ESWT. The purpose of this study was to evaluate the long-term clinical outcomes of ESWT in plantar fibromatosis confirmed by ultrasonography and to investigate the long-term ultrasonographic changes of plantar fibroma after ESWT.

## Materials and methods

### Subjects

This was a retrospective cohort study conducted at the foot clinic at Kangbuk Samsung Hospital from October 2011 to May 2018. Medical charts of 26 patients (30 feet) with plantar fibromatosis confirmed by ultrasonography were reviewed. Clinically, patients with clinical presumption of plantar fibromatosis (plantar pain accompanied by a palpable nodular lesion of the plantar fascia) were indicated for diagnostic ultrasonography. Plantar fibromatosis was confirmed when a hypoechogenic or mixed echogenic nodule with a longitudinally elongated shape embedded in the plantar fascia was found on ultrasonography (Fig 1) [6, 7, 26]. The maximal size of the plantar fibroma was measured in three dimensions: length, width, and thickness.

Finally, a total of 10 feet for plantar fibromatosis underwent ESWT in accordance with the ESWT protocol modified from previous reports [27–29] (Fig 2A). Exclusion criteria were as follows: history of trauma, calcaneal stress fracture, tarsal tunnel syndrome, systemic inflammatory disease, lumbosacral radiculopathy, other neurologic disorders of lower limb, and history of previous steroid injection. Additionally, medical charts of 144 patients (168 feet) with plantar fasciitis confirmed by ultrasonography were reviewed. Clinically, patients with clinical presumption of plantar fasciitis (heel pain with localized tenderness over the medial aspect of the calcaneal tuberosity) were indicated for diagnostic ultrasonography. Plantar fasciitis was confirmed when the plantar fascia was thicker than 4.0 mm on ultrasonography [30–33]. Subsequently, 42 feet with plantar fasciitis who underwent ESWT with the equal protocol to the feet with plantar fibromatosis were included this study for the comparison of clinical features (Fig 2B). This study was approved by the Institutional Ethics Review Board of Kangbuk Samsung Hospital, and the requirement for informed consent was waived due to retrospective study design. (KBSMC 2019-03-014) All methods were performed in accordance with the relevant guidelines and regulations. Data were analyzed anonymously.

### ESWT protocol

ESWT (0.10–0.14 mJ/mm$^2$ energy density (ED) according to patient' tolerance; 900 shocks, weekly interval) was performed when the Roles-Maudsley Score (RMS) was a "Poor" or "Fair" grade and the pain was reported to persist for more than 6 months despite conservative treatment. A maximum of 12 ESWT sessions was conducted until the RMS reached a "Good" or "Excellent" grade. When patients refused to continue the ESWT due to treatment pain or post-treatment soreness, we regarded it as treatment failure, but completed follow-ups and included the data in the results. ESWT protocol modified from previous reports was used [27–29]. The

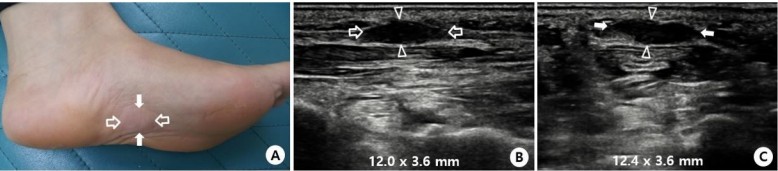

**Fig 1. Ultrasonographic diagnosis of plantar fibromatosis and measurement of plantar fibroma.** (A) Patients with clinical presumption of plantar fibromatosis (plantar pain accompanied by a palpable nodular lesion of the plantar fascia) were indicated for diagnostic ultrasonography (B) longitudinal and (C) transverse 3-12-MHz ultrasonography image of plantar fibromatosis. Plantar fibromatosis was confirmed when a hypoechogenic or mixed echogenic nodule with a longitudinally elongated shape embedded in the plantar fascia was found. The maximal size of the plantar fibroma was measured in three dimensions: length (open arrows), width (white arrows), and thickness (arrowheads).

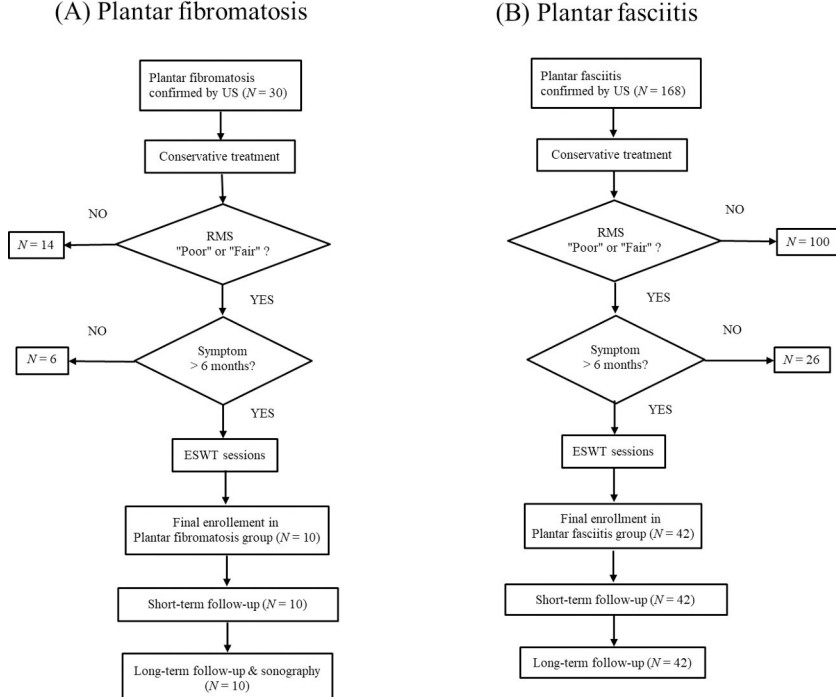

(A) Plantar fibromatosis                    (B) Plantar fasciitis

**Fig 2. Flow chart of the study.** (A) Enrollment of plantar fibromatosis group. (B) Enrollment of plantar fasciitis group US, ultrasonography; RMS, Role-Maudsley score; ESWT, extracorporeal shock wave therapy.

ESWT was applied using Evotron® (SwiTech, Kreuzlingen, Switzerland), specifically, the electrohydraulic type. The patients were stationed in the prone position, and a shock wave was applied to the tender area on the plantar fibroma in the plantar fibromatosis group or to the calcaneal insertion area of the plantar fascia in the plantar fasciitis group. All patients were recommended to reduce their activity level and avoid impact activities such as excessive walking or running etc.

## Outcome measures

The clinical outcomes were evaluated with the Numerical Rating Scale (NRS) for subjective pain and the RMS for functional outcomes. The NRS is an 11-point pain intensity rating scale, where a value of 10 points indicate worst possible pain and 0 point indicates no pain. The RMS is a subjective 4-point assessment of limitations of activity (Table 1).

In both plantar fibromatosis and plantar fasciitis groups, the NRS and RMS were assessed before each ESWT session, at short-term follow-up, and at long-term follow-up. A greater than 50% reduction in the NRS and achievement of a "good" or "excellent" grade in the RMS

**Table 1. Roles and Maudsley score.**

| Grade | Point | Interpretation |
|---|---|---|
| Excellent | 1 | No pain, full movement and activity |
| Good | 2 | Occasional discomfort, full movement and activity |
| Fair | 3 | Some discomfort after prolonged activity |
| Poor | 4 | Pain-limiting activities |

were regarded as treatment success. Short-term follow-up was accomplished one week after ESWT sessions were completed, and long-term follow-up was performed mean 34.0 months (range 11 to 63 months) after ESWT for the plantar fibromatosis group; and mean 37.7 months (range 9 to 80 months) for the plantar fasciitis group.

In the plantar fibromatosis group, follow-up ultrasonography was conducted at long-term follow-up to investigate morphologic changes of the plantar fibroma after ESWT. The maximal size of the plantar fibroma was evaluated by ultrasonography in three dimensions: length, width, and thickness (Fig 1).

## Statistical analysis

Repeated measures analysis of variance (ANOVA) and Paired t-test were used to analyze the changes of NRS and RMS. The Student t-test and Fisher's exact test were used for the comparison of demographics between two groups. The morphologic changes of the plantar fibromas were analyzed with the Paired t-test. The Pearson's chi-square test was used to compare the success rate of treatment between the two groups. All statistical analyses were performed with the IBM SPSS Statistics, version 24.0 (Armonk, NY: IBM Corp). A $P < 0.05$ were considered statistically significant.

## Results

The basic characteristics of subjects are shown in Table 2. Repeated measures ANOVA revealed a significant improvement of NRS and RMS over time in plantar fibromatosis group at short-term follow-up ($P < 0.001$) and at long-term-follow-up ($P < 0.001$). In the plantar fibromatosis group, the mean NRS was significantly reduced from 6.2 ± 1.3 (baseline) to 1.8 ± 1.0 (short-term follow-up) and 0.6 ± 1.1 (long-term follow-up). Baseline RMS (3.5 ± 0.5) were significantly improved at short-term follow-up (2.0 ± 0.8, $P < .001$) and long-term

**Table 2. Demographics and characteristics of subjects.**

| Characteristics | Plantar fibromatosis ($N = 10$) | Plantar fasciitis ($N = 42$) | P value |
|---|---|---|---|
| Age, year (range) | 49.1 ± 11.3 (36–75) | 50.6 ± 10.4 (33–76) | .691[a] |
| Gender | | | |
| male | 6 | 17 | .264[b] |
| female | 4 | 25 | |
| Duration of symptoms, month (range) | 13.0 ± 9.7 (6–36) | 11.3 ± 5.2 (6–25) | .448[a] |
| Follow-up period, month (range) | 34.0 ± 13.4 (11–63) | 37.7 ± 19.8 (9–80) | .580[a] |
| Affected site | | | |
| right | 3 | 19 | .381[b] |
| left | 7 | 23 | |
| Baseline NRS (range) | 6.2 ± 1.3 (4–8) | 5.7 ± 1.6 (3–10) | .447[a] |
| Baseline RMS (range) | 3.5 ± 0.5 (3–4) | 3.3 ± 0.4 (3–4) | .414[a] |
| Total number of ESWT (range) | 7.8 ± 2.9 (5–12) | 6.5 ± 2.8 (1–12) | .228[a] |
| Maximal size of fibroma on ultrasonography | | | |
| length, mm (range) | 13.5 ± 4.6 (9.2–22.2) | | |
| width, mm (rane) | 10.3 ± 3.5 (5.7–16.0) | | |
| thickness, mm (range) | 4.4 ± 1.0 (3.3–6.3) | | |

Abbreviations: NRS, numeric rating scale; RMS, Roles-Maudsley score; ESWT, extracorporeal shock wave therapy.

[a]Student's *t*-test

[b]Fisher's exact test.

**Table 3. Changes of NRS and RMS after ESWT in plantar fibromatosis group.**

| | Plantar fibromatosis group (N = 10) | |
| --- | --- | --- |
| | Mean ± SD (range) | P value |
| NRS | | |
| Baseline | 6.2 ± 1.3 (4.0–8.0) | |
| Short-term follow-up | 1.8 ± 1.0 (0.5–3.5) | <0.001* |
| Long-term follow-up | 0.6 ± 1.1 (0.0–3.0) | <0.001** |
| RMS | | |
| Baseline | 3.5 ± 0.5 (3.0–4.0) | |
| Short-term follow-up | 2.0 ± 0.8 (1.0–3.0) | <0.001* |
| Long-term follow-up | 1.4 ± 0.8 (1.0–3.0) | <0.001** |

Abbreviations: NRS, numeric rating score; RMS, Roles-Maudsley score; ESWT, extracorporeal shock wave therapy.

*, between baseline and short-term follow-up by paired *t*-test

**, between baseline and long-term follow-up by repeated measures ANOVA

follow-up (1.4 ± 0.8, $P$ < .001) (Table 3). With regard to plantar fasciitis group, baseline NRS (5.7 ± 1.6) and RMS (3.3 ± 0.4) were significantly improved at short-term follow-up (NRS, 2.5 ± 2.1; RMS, 2.1 ± 0.8, $P$ < .001, respectively) and at long-term follow-up (NRS, 1.1 ± 2.4; RMS, 1.6 ± 0.7, $P$ < .001, respectively) (Table 4).

The treatment success rate in fibromatosis group were as follows: seven feet (70.0%) at short-term follow-up and 8 feet (80%) at long-term follow-up achieved treatment success. In plantar fasciitis group, short-term success was achieved in 28 feet (66.7%) and long-term success was in 35 feet (83.3%). There were no significant differences in success rate between the two groups in both short-term and long-term follow-ups (Table 5).

Follow-up ultrasonography indicated that the mean thickness of fibromas was significantly reduced from baseline (4.4±1.0 mm, range 3.3–6.3) to long-term follow-up (2.6±0.8 mm, range 0.4–3.9, $P$ = 0.003). However, the mean length and width of fibromas were not significantly changed (length, $P$ = 0.207; width, $P$ = 0.090). The mean fibroma length was estimated to be 13.5±4.6 mm (range 9.2–22.2) at baseline and 12.3±4.9 mm (range 8.9–24.9) at long-term follow-up. The mean width was 10.3±3.5 (range 5.7–16.0) at baseline and 9.1±3.1 mm (range 5.4–15.3) at long-term follow-up (Fig 3). None of the cases experienced complete resolution; however, softening of the fibroma was observed in all cases.

**Table 4. Changes of NRS and RMS after ESWT in plantar fasciitis group.**

| | Plantar fasciitis group (N = 42) | |
| --- | --- | --- |
| | Mean ± SD (range) | P value |
| NRS | | |
| Baseline | 5.7 ± 1.6 (3.0–10.0) | |
| Short-term follow-up | 2.5 ± 2.1 (0.0–8.0) | <0.001* |
| Long-term follow-up | 1.1 ± 2.4 (0.0–6.0) | <0.001** |
| RMS | | |
| Baseline | 3.3 ± 0.4 (3.0–4.0) | |
| Short-term follow-up | 2.1 ± 0.8 (1.0–4.0) | <0.001* |
| Long-term follow-up | 1.6 ± 0.7 (1.0–3.0) | <0.001** |

Abbreviations: NRS, numeric rating score; RMS, Roles-Maudsley score; ESWT, extracorporeal shock wave therapy.

*, between baseline and short-term follow-up by paired *t*-test

**, between baseline and long-term follow-up by repeated measures ANOVA

**Table 5. Success rate of ESWT.**

|  | Plantar fibromatosis group (*N* = 10) | Plantar fasciitis group (*N* = 42) | *P value* |
|---|---|---|---|
| Short-term follow-up | 7 (70.0%) | 28 (66.7%) | .840* |
| Long-term follow-up | 8 (80.0%) | 35 (83.3%) | .802* |

Abbreviation: ESWT, extracorporeal shock wave therapy.

*, between plantar fibromatosis group and plantar fasciitis group by *chi*-square

## Discussion

In this study, subjective pain score and functional score were significantly improved one week after ESWT and at long-term follow-up (mean 34.0 months after ESWT) in plantar fibromatosis group. The results of our study are in accordance with two previous case series [24, 25]. Knobloch et al. [25] used two sessions of high-energy focused ESWT (1.24 mJ/mm$^2$ ED, 2000 pulse, weekly) in 6 patients with plantar fibromatosis. The mean visual analogue scale (VAS) was reduced from 6 to 2 one week after the ESWT sessions, and it decreased to 1 at 3 months of follow-up. Frizziero et al. [24] conducted 4 consecutive ESWT sessions (maximal 0.20 mJ/mm$^2$ ED, 1600 shocks, weekly) in 3 feet with plantar fibromatosis. The mean VAS was reduced from 5.6 to 0.6 at 6 months of follow-up. Foot Function Index Scores were also improved in all

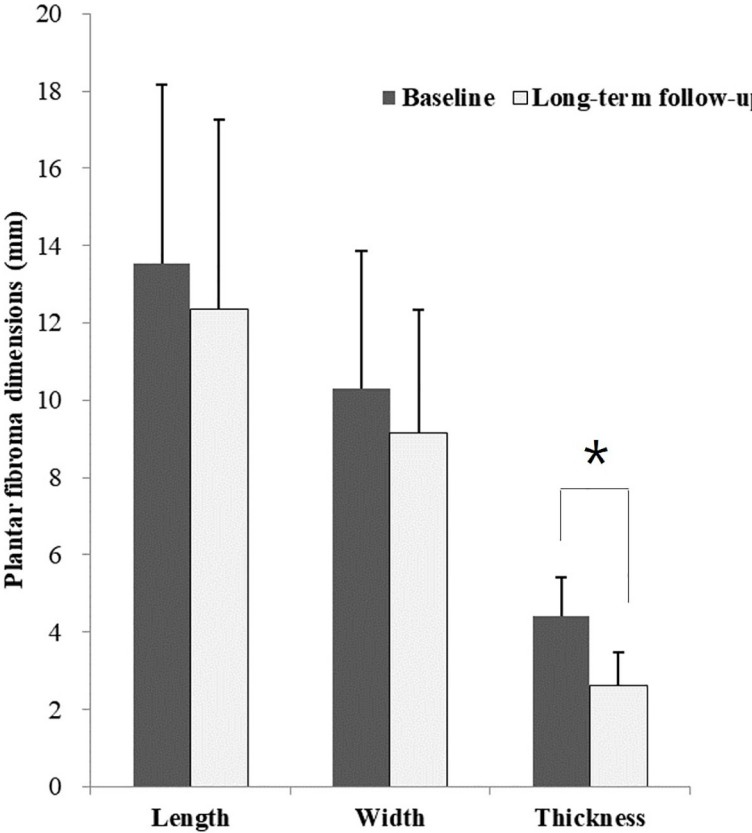

**Fig 3. Changes of plantar fibroma in follow-up ultrasonography after ESWT.** The mean thickness of fibroma was significantly reduced, while length and width were not significantly changed. ESWT, extra corporeal shock wave therapy.

3 feet. Our ESWT parameters were as follows: a maximum of 12 sessions (mean 7.8±2.9, range 1–12) with 0.10–0.14 mJ/mm$^2$ ED and 900 shocks on a weekly basis. In our study, the mean NRS was improved from 6.2 at baseline to 1.8 at short term follow-up (one week after ESWT sessions), and it decreased to 0.6 at long-term follow-up (mean 34 months after ESWT sessions). The mean RMS was also improved from 3.5 at baseline to 2.0 at short-term follow-up, and 1.4 at long-term follow-up. In addition, treatment success was achieved in 7 feet (70.0%) at short-term follow-up and 8 feet (80.0%) at long-term follow-up in the plantar fibromatosis group. These results are comparable to those of the plantar fasciitis group, in which 28 feet (66.7%) and 35 feet (83.3%) achieved treatment success at short term and long-term follow-up, respectively. Based on these findings, ESWT can be considered as a valid therapeutic option for pain relief and for functional improvement in chronic painful plantar fibromatosis, although there is variability in the protocol for treatment.

In the literature, ESWT has been shown to be effective for pain relief in penile fibromatosis (Peyronie's disease) and palmar fibromatosis (Dupuytren's disease) which is uncommon, benign, and hyperproliferative fibrous tissue disease. However, there is limited published data for plantar fibromatosis. Although the mechanisms of the analgesic effect of ESWT are unclear, hyperstimulation of nociceptors that alters cell membrane permeability of nociceptors [34], suppression of neurotransmitter substance P, and increased local pain-inhibiting substances [35] have been suggested in musculoskeletal diseases such as plantar fasciitis, Achilles tendinopathy and gluteal tendinopathy. Additionally, stimulation of nociceptors may also play a role in tendon remodeling, as it may induce release of neuropeptides, resulting in fibroblast stimulation and vasodilation [34]. In terms of fibromatosis, it is thought that ESWT stimulates biosynthesis of the extracellular matrix by tendon fibroblasts, which could help in counteracting the maturation process of myofibroblasts and lead to reduced tissue contraction [24, 36].

There was no significant morphologic change in fibromas on ultrasonography, with the exception of reduced thickness, and there was no case in which the fibroma was completely resolved after ESWT. This result is similar to previous reports on penile fibromatosis, in which ESWT is unlikely to reduce the size of the fibroma [14, 15, 23]. There are only two reports that ESWT reduced size of penile fibroma and improved penile curvature [17, 18]. However, given the progressive nature of fibroma, ESWT may have interfered with the growth of the fibroma and could have been advantageous for avoiding the need for radiation or surgery. Thus, such effects could be clinically beneficial, even if the size dose not significantly decrease. Actually, softening of fibroma after ESWT was noted in all 10 cases in this study, which was in accord with the previous reports on plantar fibroma by Frizziero et. al. [24] and Knobloch et. al. [25]. Because standardized method has not been established, we also confirmed softening of fibroma by palpation after patient's report as with the previous reports. Further studies using more objective method, such as shear wave elastography, would be needed.

No serious adverse effects were observed after the ESWT session in any of the participants. Post-treatment soreness was seen at the ESWT site in 7 feet (70.0%) in the plantar fibromatosis group and 38 feet (90.5%) in the plantar fasciitis group. However, this symptom subsided spontaneously within mean 1.0±0.9 days in the plantar fibromatosis group and 1.6±1.3 days in the plantar fasciitis group with no significant difference between the two groups.

There were several limitations to this study. First, due to the retrospective study design, there was a lack of information; therefore, the concomitant conservative treatment received, such as physical therapy, NSAIDs, stretching and use of custom-made insoles may have affected the outcomes. Second, there was no placebo control group. Thus, the effect of the natural progression of this condition and sonographic change without ESWT could not be assessed. Third, only a small number of subjects was included in this study, which could be too small to draw a definite conclusion. The small number of enrolled subjects were inevitable

because plantar fibromatosis is not a common disease. However, further study with larger number of subjects would be still needed.

## Conclusion

ESWT can have a beneficial long-term effect on pain relief and functional outcomes in painful plantar fibromatosis, but it is unlikely to affect the ultrasonographic morphology of plantar fibroma, except for reducing the thickness. These are preliminary findings, and must be confirmed in a randomized placebo control study.

## Supporting information

**S1 File.**
(XLSX)

**S2 File.**
(XLSX)

## Author Contributions

**Conceptualization:** Kyung Jae Yoon, Yong-Taek Lee.

**Data curation:** Jin Tae Hwang, Jae Hyeoung Choi, Hee-Jin Park, Yong-Taek Lee.

**Formal analysis:** Jin Tae Hwang, Jae Hyeoung Choi, Hee-Jin Park, Yong-Taek Lee.

**Funding acquisition:** Yong-Taek Lee.

**Investigation:** Jin Tae Hwang, Kyung Jae Yoon, Jae Hyeoung Choi, Yong-Taek Lee.

**Methodology:** Jin Tae Hwang, Jae Hyeoung Choi, Hee-Jin Park, Yong-Taek Lee.

**Project administration:** Kyung Jae Yoon, Yong-Taek Lee.

**Resources:** Kyung Jae Yoon, Yong-Taek Lee.

**Software:** Jin Tae Hwang, Jae Hyeoung Choi.

**Supervision:** Kyung Jae Yoon, Chul-Hyun Park, Young Sook Park, Yong-Taek Lee.

**Validation:** Kyung Jae Yoon, Chul-Hyun Park, Hee-Jin Park, Young Sook Park, Yong-Taek Lee.

**Visualization:** Jin Tae Hwang, Hee-Jin Park, Yong-Taek Lee.

**Writing – original draft:** Jin Tae Hwang, Yong-Taek Lee.

**Writing – review & editing:** Chul-Hyun Park, Young Sook Park, Yong-Taek Lee.

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
