## [Decision Letter · Decision Letter 0]

24 Jun 2020

PONE-D-20-02966

Follow-up of clinical and sonographic features after extracorporeal shock wave therapy in painful plantar fibromatosis

PLOS ONE

Dear Dr. Lee,

Thank you for submitting your manuscript to PLOS ONE. After careful consideration, we feel that it has merit but does not fully meet PLOS ONE’s publication criteria as it currently stands. Therefore, we invite you to submit a revised version of the manuscript that addresses the points raised during the review process.

The reviewers raised a number of concerns about the methodological approach used in the study. They specifically felt that there were issues with the small sample size that need to be addressed. The full reviewer comments can be found below.

We look forward to receiving your revised manuscript.

Kind regards,

Natasha McDonald

Associate Editor

PLOS ONE

Journal Requirements:

Additional Editor Comments (if provided):

Reviewers' comments:

Reviewer's Responses to Questions

**Comments to the Author**

1. Is the manuscript technically sound, and do the data support the conclusions?

Reviewer #1: Yes

Reviewer #2: Partly

2. Has the statistical analysis been performed appropriately and rigorously? 

Reviewer #1: Yes

Reviewer #2: N/A

3. Have the authors made all data underlying the findings in their manuscript fully available?

Reviewer #1: Yes

Reviewer #2: Yes

4. Is the manuscript presented in an intelligible fashion and written in standard English?

Reviewer #1: Yes

Reviewer #2: Yes

5. Review Comments to the Author

Reviewer #1: This study was to evaluate the long-term clinical outcome of ESWT in the plantar fibromatosis and ultrasonographic changes. As your result, low-energy ESWT can have a beneficial long-term effect on pain relief and functional outcome. I agree this article is appropriate to publish. However, this is a retrospective study, there are some limitation. The adjuvant treatments affected the outcome and lack of placebo control group.

Reviewer #2: This is a retrospective study investigating the clinical and sonographic features of plantar fibromatosis after ESWT.

General comments:

1. As a retrospective study, the number of enrolled subjects (feet) were too small to draw a conclusion.

2. In addition, there is lack of comparison group for sonographic change in plantar fibroma without ESWT.

3. A maximum of 12 sessions of ESWT received in the enrolled participants. What factors may be possibly related to treatment success after less sessions?

4. Although the thickness of plantar fibroma is reduced in long-term follow-up period, how did the authors confirm the softening of the fibroma, as stated in page 11, line 196& page 13, line 246-247?

Specific comments:

1. The authors stated the low energy ESWT in the text. The definition of low-energy ESWT should be clearly stated in the text

2. In medical chart review, there is a lack of information on the concomitant conservative treatments received

3. In page 6, line 123-124, the authors stated “A maximum of 12 ESWT sessions was conducted until the RMS reached a “Good”or “Excellent” grade or patients refused to continue the ESWT”. Possible information on the reasons of refuse further ESWT? Were the data included anyway if patients refused to continue ESWT?

4. Table 3 and Figure 3 are stating the same results from repeated measures ANOVA.

6. PLOS authors have the option to publish the peer review history of their article (what does this mean?). If published, this will include your full peer review and any attached files.

Reviewer #1: No

Reviewer #2: No

---

## [Author Response · Author response to Decision Letter 0]

4 Jul 2020

Dear Editor-in-Chief

Thanks for your reviews and comments for our manuscript. We tried to response reviewer’s comment properly and clarify the text when needed. We hope that the reviewers will find our responses to their comments satisfactory, and our manuscript could be accepted and then published in your journal as soon as possible. We look forward to hearing from you soon.

<Author’s comment>

Please find our response (in blue) to reviewer’s specific comments (in black) below. The corrected sentences are highlighted with yellow color. 

Reviewer #1

 This study was to evaluate the long-term clinical outcome of ESWT in the plantar fibromatosis and ultrasonographic changes. As your result, low-energy ESWT can have a beneficial long-term effect on pain relief and functional outcome. I agree this article is appropriate to publish. However, this is a retrospective study, there are some limitation. The adjuvant treatments affected the outcome and lack of placebo control group.

-> Thanks for your reviews and comments. We totally agree with your comments that the adjuvant treatments affected the outcome and lack of placebo control group. We suppose that this factor could be compensated to some degree because we confined the subjects to the feet with chronic intractable painful plantar fibromatosis whose symptom did not improve after conventional conservative treatment and lasted more than 6 months. However, placebo control group would be still needed to assess the pure effect of the ESWT. Therefore, we addressed these issues in limitation section as follows: 

There were several limitations to this study. First, due to the retrospective study design, there was a lack of information; therefore, the concomitant conservative treatment received, such as physical therapy, NSAIDs, stretching and use of custom-made insoles may have affected the outcomes. Second, there was no placebo control group. Thus, the effect of the natural progression of this condition and sonographic change without ESWT could not be assessed. 

Reviewer #2: This is a retrospective study investigating the clinical and sonographic features of plantar fibromatosis after ESWT.

General comments

1. As a retrospective study, the number of enrolled subjects (feet) were too small to draw a conclusion.

-> We totally agree with reviewer’s comments that the number of enrolled subjects (feet) were too small to draw a conclusion. Therefore, we addressed these issues in limitation section as follows: 

Third, only a small number of subjects was included in this study, which could be too small to draw a definite conclusion. The small number of enrolled subjects were inevitable because plantar fibromatosis is not a common disease and our strict inclusion criteria confined subjects to the feet with chronic intractable painful plantar fibromatosis whose symptom did not improve after conventional conservative treatment and lasted more than 6 months. However, further study with larger number of subjects would be still needed.

2. In addition, there is lack of comparison group for sonographic change in plantar fibroma without ESWT.

-> We agree with the reviewer’s comment the there was no control group for sonographic change in plantar fibroma without ESWT. Thus, we addressed this issue in our limitation section as follows: 

Second, there was no placebo control group. Thus, the effect of the natural progression of this condition and sonographic change without ESWT could not be assessed. 

3. A maximum of 12 sessions of ESWT received in the enrolled participants. What factors may be possibly related to treatment success after less sessions?

-> Lower initial NRS, duration of onset, and post-treatment soreness was thought to be related to the less sessions, we could not found relationship among these factors in our study. We suppose further study with lager number of subjects would be needed. 

4. Although the thickness of plantar fibroma is reduced in long-term follow-up period, how did the authors confirm the softening of the fibroma, as stated in page 11, line 196& page 13, line 246-247?

-> In the previous reports on softening of plantar fibroma by Knobloch et al. and Frizziero et al., softening of fibroma was confirmed by palpation after patient’s report. Because standardized method has not been established, we also confirmed softening of fibroma by palpation after patient’s report as with the previous reports. Therefore, we addressed this issue in our manuscript as follows: 

Actually, softening of fibroma after ESWT was noted in all 10 cases in this study, which was in accord with the preveious reports on plantar fibroma by Frizziero et. al. [24] and Knobloch et. al. [25]. Because standardized method has not been established, we also confirmed softening of fibroma by palpation after patient’s report as with the previous reports. Further studies using more objective method, such as shear wave elastogrphy, would be needed.

Specific comments

1. The authors stated the low energy ESWT in the text. The definition of low-energy ESWT should be clearly stated in the text

-> Shock-wave therapy is usually classified according to the energy flux density administered as follows: low-energy (< 0.08-0.27 mJ/mm2), medium-energy (0.28–0.59 mJ/mm2), and high-energy (> 0.6 mJ/mm2); or low-energy (< 0.12 mJ/mm2) and high-energy (> 0.12 mJ/mm2). [J Bone Joint Surg Br 2004;86-B:165-71] While these terms ‘high’, ‘medium’ and ‘low’ energy are commonly used in the literature, there is no clear consensus on the threshold values. Therefore, we erased the term ‘low-energy’ from our conclusion section. 

2. In medical chart review, there is a lack of information on the concomitant conservative treatments received

-> We addressed these issues in limitation section as follow: 

There were several limitations to this study. First, due to the retrospective study design, there was a lack of information; therefore, the concomitant conservative treatment received, such as physical therapy, NSAIDs, stretching and use of custom-made insoles may have affected the outcomes. 

3. In page 6, line 123-124, the authors stated “A maximum of 12 ESWT sessions was conducted until the RMS reached a “Good”or “Excellent” grade or patients refused to continue the ESWT”. Possible information on the reasons of refuse further ESWT? Were the data included anyway if patients refused to continue ESWT?

-> Patients refused to continue ESWT when they felt that there was no effect due to treatment pain or post-treatment soreness. When the patients refused ESWT in the middle of ESWT sessions, we regarded it as treatment failure, but completed follow-ups and included data in the results. We addressed this issue in our manuscript as follows: 

A maximum of 12 ESWT sessions was conducted until the RMS reached a “Good” or “Excellent” grade. When patients refused to continue the ESWT due to treatment pain or post-treatment soreness, we regarded it as treatment failure, but completed follow-ups and included the data in the results.

4. Table 3 and Figure 3 are stating the same results from repeated measures ANOVA.

-> We agree with your comments. Therefore, we deleted the Figure 3 in our manuscript.

---

## [Editor Report · Decision Letter 1]

28 Jul 2020

Follow-up of clinical and sonographic features after extracorporeal shock wave therapy in painful plantar fibromatosis

PONE-D-20-02966R1

Dear Dr. Lee,

We’re pleased to inform you that your manuscript has been judged scientifically suitable for publication and will be formally accepted for publication once it meets all outstanding technical requirements.

Kind regards,

Ezio Lanza, M.D.

Academic Editor

PLOS ONE

Additional Editor Comments (optional):

Dear Authors,

The paper shows sufficient quality to grant publication.

Please remove claims of primacy at lines 239-240 "To our knowledge, this is the first study to investigate the morphologic changes after ESWT in plantar fibromatosis. " which do not add value to the discussion.

Also, avoid repeating inclusion criteria at 265-268. Stating that plantar fibromatosis is an uncommon disease, it's enough of an explanation for the small cohort.

For better readability, please invert the paragraph order in conclusion. First, state that "ESWT can have a beneficial long-term effect on pain relief and functional outcomes in painful plantar fibromatosis.", but "ESWT is unlikely to affect the ultrasonographic morphology of plantar fibroma, except for reducing the thickness." then complete saying that "these are preliminary findings, and must be confirmed in a randomized placebo control study.